# The Effect of Polydeoxyribonucleotide Extracted from Salmon Sperm on the Restoration of Bisphosphonate-Related Osteonecrosis of the Jaw

**DOI:** 10.3390/md17010051

**Published:** 2019-01-11

**Authors:** Deok-Won Lee, Hoon Hyun, Sungsu Lee, So Yeon Kim, Gyu-Tae Kim, Sewook Um, Sung Ok Hong, Heung Jae Chun, Dae Hyeok Yang

**Affiliations:** 1Department of Oral & Maxillofacial Surgery, Kyung Hee University Dental Hospital at Gangdong, Kyung Hee University, Seoul 153-782, Korea; verycutebear@hanmail.net; 2Department of Biomedical Sciences, Chonnam National University Medical School, Gwangju 61469, Korea; hhyun@chonnam.ac.kr; 3Department of Otolaryngology-Head and Neck Surgery, Chonnam National University Medical School, Gwangju 61469, Korea; minsunglss@naver.com; 4Department of Dental Hygiene, College of Health Sciences, Cheongju University, Cheongju 28503, Korea; goodany00@gmail.com; 5Department of Oral and Maxillofacial Radiology, Kyung Hee University, Seoul 02447, Korea; latinum.omfr@khu.ac.kr; 6Department of Veterinary Surgery, College of Veterinary Medicine, Seoul National University, Seoul 08826, Korea; umsewook@gmail.com; 7Department of Dentistry, Catholic Kwandong University, School of Medicine, International St. Mary’s Hospital, Incheon 22711, Korea; cathead81@naver.com; 8Department of Biomedical Sciences, College of Medicine, The Catholic University of Korea, Seoul 06591, Korea; chunhj@catholic.ac.kr; 9Institute of Cell and Tissue Engineering, College of Medicine, The Catholic University of Korea, Seoul 06591, Korea

**Keywords:** bisphosphonate, bisphosphonate-related osteonecrosis of the jaw, polydeoxyribonucleotide, necrotic bone percentages, blood vessels, osteoclasts

## Abstract

Bisphosphonates (BPs) used for treating skeletal diseases can induce bisphosphonate-related osteonecrosis of the jaw (BRONJ). Despite much effort, effective remedies are yet to be established. In the present study, we investigated the feasibility of polydeoxyribonucleotide (PDRN) extracted from salmon sperm for the treatment of BRONJ, in a BRONJ-induced rat model. Compared with BRONJ-induced samples, PDRN-treated samples exhibited lower necrotic bone percentages and increased numbers of blood vessels and attached osteoclast production. Moreover, local administration of PDRN at a high concentration (8 mg/kg) remarkably resolved the osteonecrosis. Findings from this study suggest that local administration of PDRN at a specific concentration may be considered clinically for the management of BRONJ.

## 1. Introduction

Bisphosphonates (BPs) are one of the most widely used bone resorption inhibitor drugs for the management of skeletal diseases such as osteoporosis, Paget’s disease, multiple myeloma, and complications from metastatic malignancy [1]. Despite the therapeutic effect of BPs on osteoporosis, their use creates challenging problems and potentially severe side effects, including bisphosphonate-related osteonecrosis of the jaw (BRONJ) [2].

BRONJ involves bone necrosis in the maxillofacial region. The risk of BRONJ increases after BP has been used for longer than eight weeks in patients, either as previous or current treatment. General symptoms of BRONJ include soft tissue swelling, ulceration, pain, infection, and the presence of necrotic bone [3]. In most cases, BRONJ occurs in patients after dental treatment, such as tooth extraction [3]. BRONJ is considered a severe social problem, because BPs are commonly prescribed for osteoporosis in the older generations, which may adversely affect the outcome of their dental treatment.

The exact pathogenesis of BRONJ remains uncertain. Various hypotheses have been proposed, including over-suppression of osteoclastic bone resorption and bone turnover, suppression of angiogenesis, oral bacterial infection, and oral epithelial toxicity [4,5,6]. A greater understanding of BRONJ is one of the most fundamental factors for disease treatment. DNA-derived drugs have exhibited positive effects with respect to accelerating wound healing by controlling the healing mechanism [7,8,9]. Mucosal wound healing is known to be an important factor for BRONJ recovery [4,5,6]. Polydeoxyribonucleotides (PDRNs) extracted from salmon sperm, a family of DNA-derived drugs with molecular weights between 50 and 1500 KDa, may provide a good strategy for inhibiting BRONJ, due to their biological properties. PDRNs have clinically been used for wound healing in the skin and diabetic foot ulcers because of their reparatory, anti-ischemic, and anti-inflammatory properties [7,10,11]. PDRNs are also known to improve granulation tissue formation and increase angiogenesis [12]. So, we hypothesized that local treatment using PDRN in socket defects after tooth extractions inhibit the occurrence of BRONJ in ZA-treated rats, and investigated the effect of PDRN on BRONJ in this study.

We demonstrated the effect of PDRN by examining necrotic bone, blood vessels, osteoclast numbers, osteoclast surfaces, empty lacunae, attached/detached osteoclast numbers, and bone volume in a BRONJ-induced rat model. Our results suggest that a specific concentration of PDRN is effective for the treatment of BRONJ.

## 2. Results

### 2.1. Establishment of a BRONJ Rat Model and Gross Appearance of Tooth-Extracted Sites

Figure 1 shows the experimental procedure for establishing a BRONJ rat model and the H&E-stained images of the control and experimental groups. For the BRONJ model, ovariectomy was carried out in normal rats, followed by tooth extractions. Zoledronic acid (ZA; 0.6 mg/mL; Sigma-Adrich, St. Louis, MO, USA) was intraperitoneally administered twice a week for 18 weeks following tooth extraction for each rat (Figure 1A). Figure 1B shows the gross appearances of the tooth extraction sites (maxillae) in the Sham (control) vs. BRONJ-induced rats. PDRN was used to treat the BRONJ-induced rats twice a week for 20 days at doses of 2, 4, and 8 mg/kg.

Our results showed that full mucosal coverage of the defects was observed in the control group. In contrast, osteonecrosis with a yellowish-brown color due to the occurrence of BRONJ was observed in the BRONJ-induced group. Treatment with PDRN in the BRONJ-induced rats seemed to reduce the severity of the osteonecrosis.

### 2.2. Morphological Changes in Tooth-Extracted Sites

Morphological changes in the defects of the control, BRONJ-induced, and PDRN-treated rats were investigated through H&E staining (Figure 2A). After treatment with ZA every other day for 20 days, the control group (Sham) exhibited full mucosal coverage with lamella bone (LB), woven bone (WB), connective tissue (CT) and epithelial tissue (ET) in the extraction sites. In contrast, the BRONJ-induced samples showed incomplete mucosal coverage, along with fragmented connective tissue. The control group also exhibited normal osteogenesis at the site with no inflammatory infiltration, as seen in connective tissues of the extraction sites in BRONJ-induced rats. Similar to the control, PDRN-treated samples demonstrated resolution of BRONJ. In particular, a local injection of PDRN of more than 4 mg/kg resulted in full mucosal coverage. In contrast, administration of ZA resulted in considerable necrotic cortical bone with empty osteocytic lacunae (Figure 2B). Moreover, local treatment with PDRN at the defect sites of BRONJ-induced rats decreased necrotic bone formation.

### 2.3. Osteonecrotic Formation in Tooth-Extracted Sites

To further investigate osteonecrotic bone formation in the defect sites of each sample, histological evaluation of empty osteocytic lacunae was conducted (Figure 3A,B). The BRONJ-induced samples had a larger number of empty lacunae than the control group, due to increased osteonecrosis as a result of the ZA treatment. Compared with the BRONJ-induced samples, local treatment with PDRN decreased the number of empty lacunae. PDRN-treated samples exhibited a 2-fold lower percentage of necrotic bone than the BRONJ-induced samples.

### 2.4. Size and Number of Blood Vessels Formed in Tooth Extracted Sites

Figure 4 shows the size and number of blood vessels in newly formed bone in the control, BRONJ-induced, and PDRN-treated samples. Large-sized blood vessels were observed in the control and PDRN-treated samples (Figure 4A). Moreover, the PDRN-treated samples exhibited an increased number of blood vessels compared to the control and BRONJ-induced samples.

### 2.5. Number and Behavior of Osteoclasts in Tooth Extracted Sites

Tartrate-resistant acid phosphatase (TRAP) staining was conducted to evaluate the number of osteoclasts at the extraction sites (Figure 5A,B). BRONJ-induced samples exhibited the smallest number of osteoclasts compared to the control and PDRN-treated samples. In the PDRN-treated samples, the PDRN concentration contributed to the increase of osteoclasts. Specifically, a treatment of 8 mg/kg PDRN remarkably increased osteoclast numbers. On average, the number of osteoclasts in 8 mg/kg PDRN-treated samples was 3.6, 2.9, and 1.8 times larger than the BRONJ-induced, 2 mg/kg PDRN-treated, and 4 mg/kg PDRN-treated samples, respectively. The numbers of attached and detached osteoclasts were further investigated (Figure 6A,B). In the control, most of the osteoclasts were attached to the bone surfaces, indicating active bone remodeling. On the contrary, osteoclasts were predominantly detached from the bone surfaces in the test models. After PDRN treatment, the number of attached osteoclasts increased, while the number of detached osteoclasts decreased. In addition, even though the number of detached osteoclasts in the samples had no significance, the number of osteoclasts attached to the bone surfaces gradually increased with increasing PDRN concentrations. Furthermore, the number of attached osteoclasts in the 8 mg/kg PDRN-treated samples was similar to the control samples.

### 2.6. Bone Formation in Tooth Extracted Sites

Ex-vivo micro CT evaluation was carried out to confirm bone formation in the sockets after tooth extractions (Figure 7A–C). The control (Sham) rats exhibited nearly complete healing of the tooth extraction sockets. On the other hand, incomplete bone formation was observed in the BRONJ animals. Interestingly, the PDRN-treated animals demonstrated recovery bone remodeling in the sockets.

To quantitatively examine bone formation, the bone volume (BV) and bone volume/tissue volume (BV/TV) of each group was calculated (Figure 7D,E). Comparing the BRONJ models, local treatment with PDRN increased BV and BV/TV in the sockets. BV and BV/TV gradually increased as a function of the PDRN concentration. In particular, treatment with 8 mg/kg PDRN demonstrated similar BV and BV/TV to the control. This indicated that local treatment using PDRN at an appropriate concentration is effective against bone necrosis in a BRONJ-like rat model.

## 3. Discussion

The administration of BPs for osteoporosis treatment often induces BRONJ because of their effects on osteocytes, through accumulation of alveolar bone in the jaw. For example, BPs inhibit angiogenesis by obstructing the migration of neutrophils, macrophages, and osteoclast progenitor cells in tooth extraction sites [5]. BPs also increases osteonecrosis by restricting mucosal healing and providing protection against bacterial infections, due to the inhibition of granulation tissue formation [13]. Due to the increased incidence of BRONJ, many studies have attempted to understand its pathogenesis and inhibit its occurrence. However, the cause remains unclear, and successful treatments are yet to be reported. Ikebe et al. [5] suggested two possible causes for BRONJ: (1) bone turnover rate, and (2) vulnerability of the jaw to bacterial infections. Appropriate methods to control bone turnover rates and promote complete mucosal coverage of defect sites are needed.

Only rat model has been widely used as BRONJ animal model because the animal has some merits of ease of care and handling, high reproduction, completed genome mapping. Therefore, many researchers have used the animal model as BRONJ model [14]. We also selected rat model for this study. The effectiveness may be related to PDRN’s ability to induce complete mucosal healing and increase osteoclastic activity, through improvement of connective tissue formation and angiogenesis.

Among inflammatory cells, macrophages play an important role in the overall phases of wound healing, host defense, promotion and resolution of inflammation, removal of apoptotic cells, and support of cell proliferation and tissue recovery after injury [15]. Hence, for successful wound healing, appropriate function of the inflammatory cells is required in each phase [16]. Compared to the control and PDRN-treated samples, numerous inflammatory cells were observed in the BRONJ samples (Figure 2). This is likely attributed to bacterial infection due to incomplete epithelial coverage. On the other hand, the control and PDRN-treated samples exhibited a small number of inflammatory cells. This indicated that PDRN led to proliferation and remodeling of the bone, after host defense mechanisms were triggered.

Angiogenesis and normal vascularization are essential factors for improving wound healing and tissue homeostasis, respectively. Vascular endothelial growth factor (VEGF) is an inducer of angiogenesis due to its highly specific mitogen for endothelial cells [17]. However, ZA, one of most widely used BPs, reduces the mRNA and protein expressions of VEGF and decreases serum levels of VEGF and other cytokines, such as interleukin-17, involved in angiogenesis [18]. In PDRN treatment for diabetes and burns, VEGF is produced by the actions on adenosis A2 receptors [19], which may contribute to increased angiogenesis. As expected, larger sized blood vessels were observed in the PDRN-treated samples compared to the BRONJ samples, due to this PDRN effect (Figure 3A,B).

The loss of osteoclastic activity is likely to impair osteoblastic activity, causing irregular bone turnover, remodeling, and osteonecrosis [5]. Local treatment with PDRN may restore the activity of impaired osteoclasts, resulting in the resorption of necrotic bone and formation of new bone. Our results support these suggestions. In this study, we observed the involvement of attached osteoclasts in bone remodeling in PDRN-treated samples; in particular, the samples treated with 8 mg/kg PDRN. These histological results corresponded well to the results analyzed by micro CT (Figure 4). Newly formed bone was remarkably observed in PDRN-treated samples. Among the PDRN concentrations, a local injection of 8 mg/kg PDRN resulted in superior new bone formation.

The proper balance between osteoblast and osteoclast functions is vital in bone remodeling. Previous studies reported that the bone remodeling rate of cortical bone in the jaw is several times higher than in the cortex of the iliac crest of humans and in the tibial cortex in dogs [5,20,21,22]. This bone remodeling is thought to be related to the bone turnover rate. BPs have a very high affinity for hydroxyapatite crystals in bony architecture through ionic interactions between two phosphoryl groups and calcium ions (Ca^2+^). Therefore, they preferentially transfer to active bone remodeling sites and accelerate bone turnover [23,24]. BPs are known to inhibit mineral dissolution by specifically targeting farnesyl diphosphate synthase (FPP synthase) in osteoclasts. The functions of FPP synthase enzymes include regulation of cytoskeletal arrangement, vesicular trafficking, and membrane ruffling engagement in bone resorption. BPs inhibit FPP synthase activity, thereby preventing the activities of osteoclasts that destroy bones.

## 4. Materials and Methods

### 4.1. In Vivo Animal Study

New Zealand White rats (about 4 kg, n = 40) were used for evaluating the effect of PDRN (Pharma Research Products, Seongnam, Korea) on BRONJ. To induce osteoporosis, the ovary and uterus of each rat were removed by peritoneotomy, and the rats were reared for 2 months. Then, 3-aminopropionitrile fumarate salt solution (15 g/3.5 L; Sigma-Aldrich, St. Louis, MO, USA) was dissolved in the drinking water. After 15 days of this treatment, the first and second teeth in the left maxillary bone were extracted using a dental explorer. After the extraction, ZA (0.6 mL) was intraperitoneally administered to each BRONJ-induced rat for 18 weeks, twice a week. Additionally, dexamethasone (5 mg/kg) was administrated for the last two weeks, twice a week, because the drug accelerates the occurrence of BRONJ. Afterwards, three kinds of specific concentrations of PDRN (2 mg/kg, 4 mg/kg and 8 mg/kg; Rejuvenex^®^, chain lengths ranging from 50 bp to 2000 bp, PHARMARESEARCH PRODUCTS, Seongnam, Korea) were directly injected to soft tissues near to the defected sites. Six sections on each sample were prepared to investigate the effect of PDRN on BRONJ therapy both macroscopically and histologically.

### 4.2. In Vivo Animal Study Approval

The animal study was approved by the Institutional Animal Care and Use Committee (IACUC) of Kyung Hee University (KHUASP(SE)-16-063).

### 4.3. Histological Evaluations

The harvested specimens were fixed in 4% paraformaldehyde at 4 °C for 48 h, and then decalcified in a 10% EDTA solution at room temperature (RT), for 4 to 6 weeks. After dehydration with gradient ethanol, the specimens were degreased in xylene and then embedded in paraffin. Sections (5 μm thick) were analyzed histochemically with hematoxylin-eosin (H&E) and Masson’s trichrome (MT) staining according to the manufacturer’s instructions. Osteoclasts were identified by tartrate resistant acid phosphatase (TRAP) staining using an acid phosphatase leukocyte kit (Sigma-Adrich, St. Louis, MO, USA), following the protocols recommended by the manufacturer. Five uniformly spaced H&E-stained slides of the extraction sites from each specimen were scanned digitally with the ScanScope slide scanner (Nikon Instruments Inc.; Melville, NY, USA). An area of interest (~4.0 mm^2^) located within 2.0 mm of the second molar was selected. The necrotic bone, defined as any region that contained three or more empty lacunae per 1000 μm^2^, was marked. The number of empty lacunae was determined by calculating its number per 1 mm^2^. The total areas of necrotic bone were analyzed for each slide, using the ImageScope software. An average value of the necrotic bone area was calculated using five slides per rat. The percentage of necrotic bone area over total bone area was also calculated. The number and surface of osteoclasts were determined by examining TRAP activity as TRAP-positive multinucleated cells. They were calculated as osteoclast number per bone surface perimeter and as a percentage of osteoclast perimeter to bone surface perimeter, respectively.

### 4.4. Micro CT Analysis

The harvested specimens were scanned and reconstructed with SkyScan 1173 (Bruker-CT; Kontich, Belgium). Measurements were calculated using SkyScan 1173 control software (Ver 1.6; Bruker-CT; Kontich, Belgium), and carried out under 90 kVp and 88 μA, using 1 mm aluminum filtering. High resolution images were acquired at an exposure time of 500 ms, image quality of 2240 × 2240 pixels with a pixel size of 18.11 μm, and a rotation of 180°. Cross-sectional images were reconstructed using Nrecon (Ver 1.6.10.4; Bruker-CT; Kontich, Belgium). A CT analyzer (Ver 1.14.4.1; Bruker-CT; Kontich, Belgium) and Dataviewer (Ver. 1.5.1.2; Bruker-CT; Kontich, Belgium) were employed to analyze BV of tooth-extracted sites and BV/TV (tooth-extracted sites/alveolar bone excluding the extraction sites) and to align the acquired images, respectively.

### 4.5. Statistical Analysis

All quantitative data were expressed as the mean ± standard deviation. One-way analysis of variance (ANOVA) using SPSS software (SPSS Inc., Chicago, IL, USA) was used for statistical analysis. A value of ^#^
*p* < 0.05 was considered statistically significant.

## 5. Conclusions

This study demonstrated the effect of a local PDRN injection on BRONJ treatment by modifying the occurrence of angiogenesis, the completion of full mucosal coverage, and the migration and activity of osteoclasts in defect sockets. Furthermore, it was found that the therapeutic effect of PDRN is concentration-dependent. Consequently, a local injection of PDRN may potentially be an effective therapeutic modality for BRONJ treatment, because it has been clinically shown to be effective in soft tissue regeneration.

## Figures and Tables

**Figure 1 marinedrugs-17-00051-f001:**
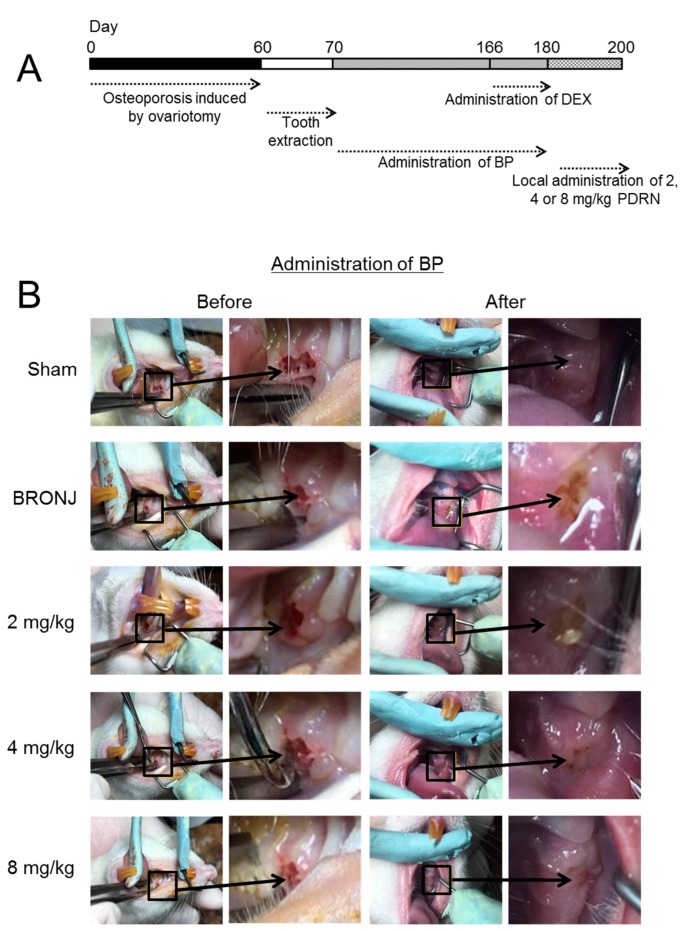
Bisphosphonate-related osteonecrosis of the jaw (BRONJ) rat model is made by the intraperitoneal administration of zoledronic acid (ZA) and is restored by a local injection of polydeoxyribonucleotide (PDRN). (**A**) The ZA was intraperitoneally administered after tooth extraction of ovary- and uterus-removed rats for 18 weeks twice a week. (**B**) For investigating the effect of PDRN on BRONJ restoration, total 10 groups were determined (n = 5~6 rats per group). Black opened squares and arrows indicate tooth extracted sites.

**Figure 2 marinedrugs-17-00051-f002:**
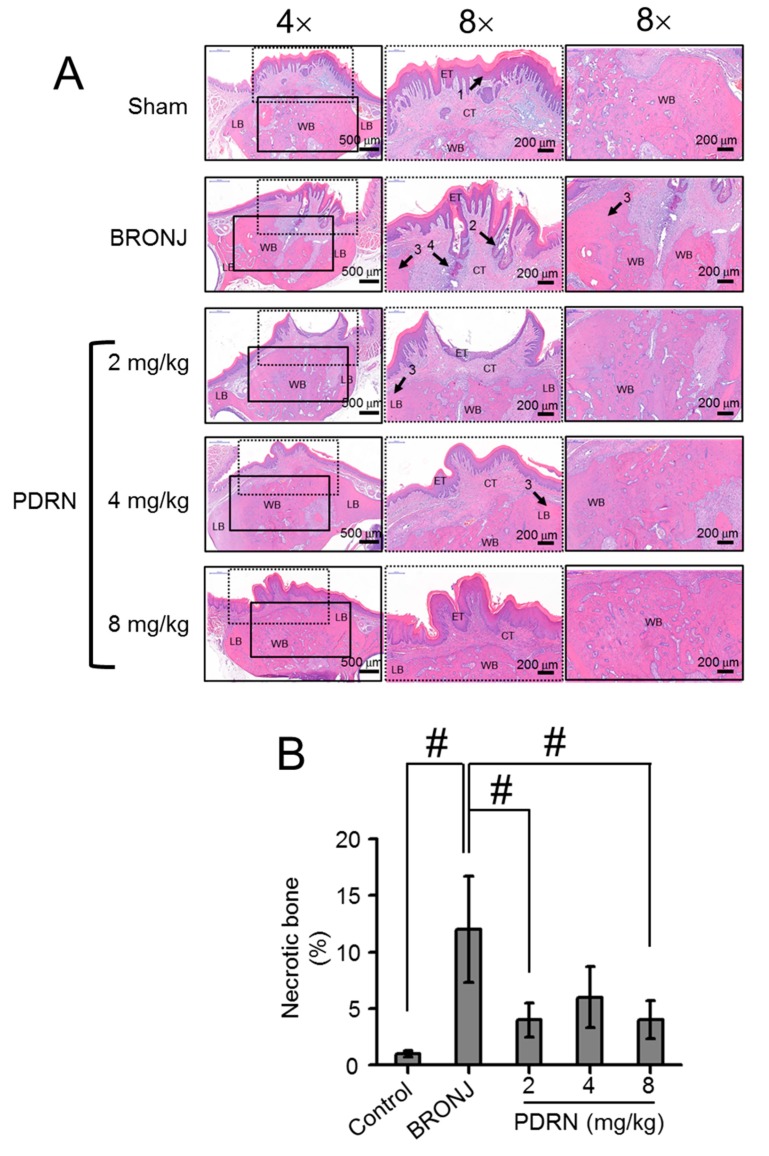
Local injection of PDRN restores BRONJ by improving full mucosal coverage and decreasing necrotic bone formation. (**A**) Tissues around tooth extracted sites in control (Sharm), BRONJ-induced, and three concentrations of PDRN (2 mg/kg, 4 mg/kg and 8 mg/kg) treated rats were stained H&E and the morphological changes were observed at magnifications of 4× and 8×. The scale bars at 4× and 8× indicates 500 μm and 200 μm, respectively. Black and black dotted squares indicate SEM images magnified at 8×. The 1, 2, 3, and 4 of arrows indicate normal ephithelial lining, ephithelial migration, necrotic bone, and inflammatory infiltrate, respectively. (**B**) Percentage of necrotic bone was calculated using ImageScope software. This data were expressed as the mean ± standard deviation. Statistical analysis was carried out using one-way analysis of variance (ANOVA) (n = 6 rats per group; ^#^
*p* < 0.05). The results of PDRN-treated groups were compared to those of control and BRONJ-induced groups.

**Figure 3 marinedrugs-17-00051-f003:**
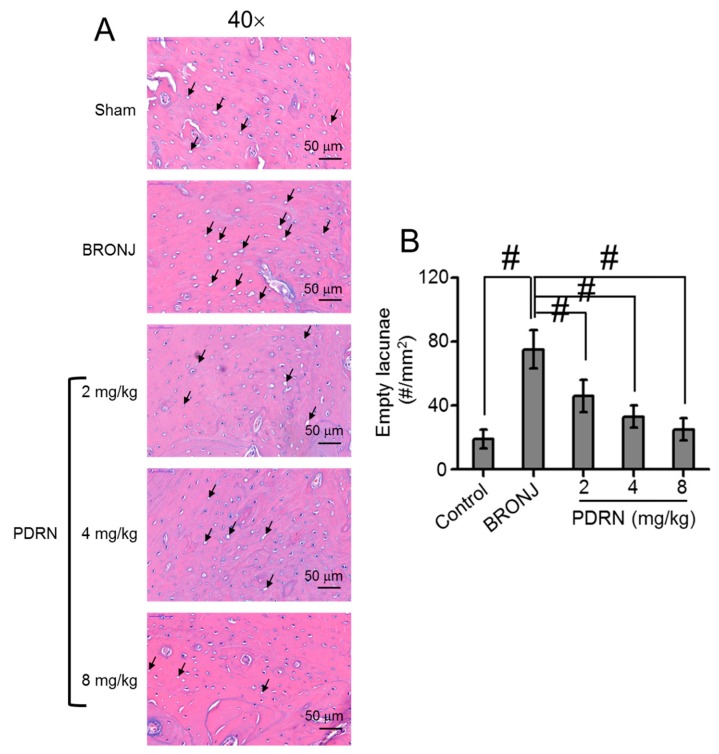
Effect of local PDRN injection on the number of empty lacunae (#/mm^2^) at tooth extracted sites. (**A**) Representative H&E stained images around tooth extracted sites of control (Sharm), BRONJ-induced, and three concentrations of PDRN (2 mg/kg, 4 mg/kg, and 8 mg/kg) treated rats after animal test for 280 days. The empty lacunae were observed at a magnification of 40×, which were expressed as black arrows. The scale bar indicates 50 μm. (**B**) Empty lacunae were measured by calculating the number per mm^2^. Statistical analysis was carried out using one-way analysis of variance (ANOVA) (n = 6 rats per group; ^#^
*p* < 0.05). The results of PDRN-treated groups were compared to those of control and BRONJ-induced groups.

**Figure 4 marinedrugs-17-00051-f004:**
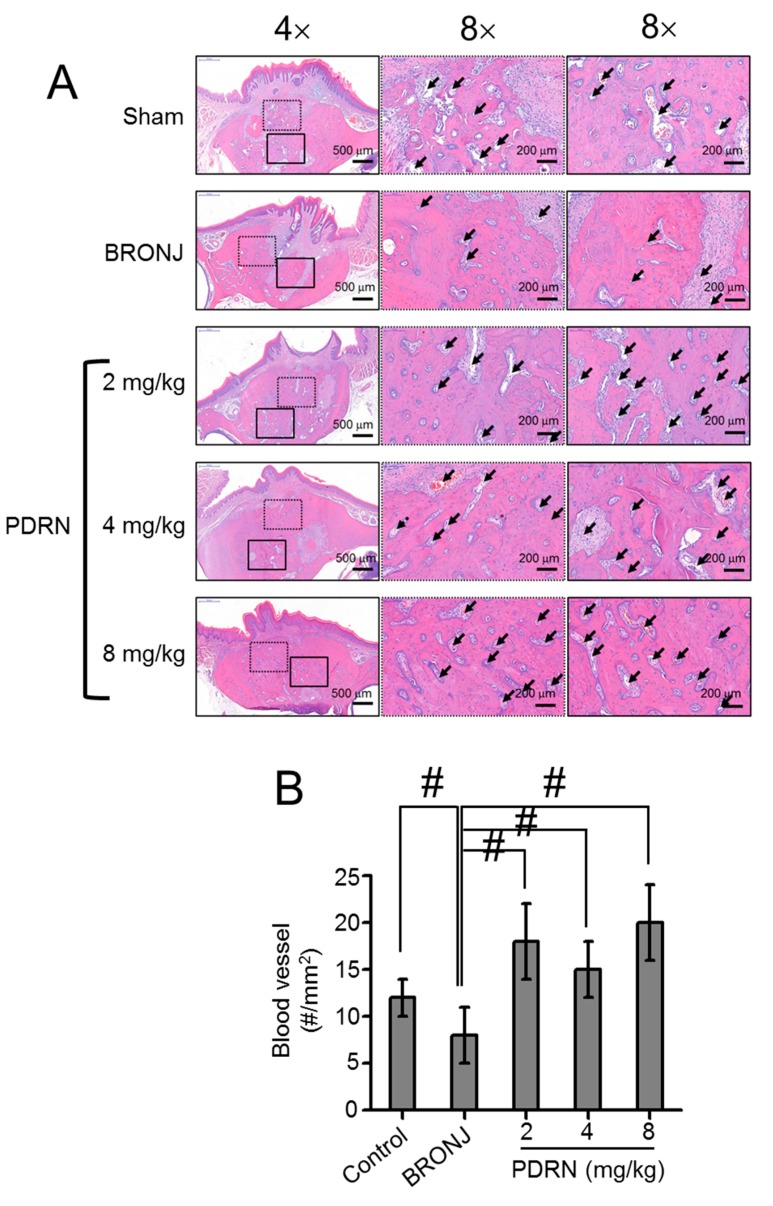
Local injection of PDRN restores BRONJ by improving new blood vessel formation. (**A**) Tissues around tooth extracted sites in control (Sharm), BRONJ-induced, and three concentrations of PDRN (2 mg/kg, 4 mg/kg and 8 mg/kg) treated rats were stained H&E and the size and number of newly formed blood vessels were observed at magnifications of 4× and 8×. The scale bars at 4× and 8× indicates 500 μm and 200 μm, respectively. Black block dotted squares indicate SEM images magnified at 8×. Black arrows indicate blood vessels. (**B**) Blood vessels were measured by calculating the number per mm^2^. Statistical analysis was carried out using one-way analysis of variance (ANOVA) (n = 6 rats per group; ^#^
*p* < 0.05). The results of PDRN-treated groups were compared to those of control and BRONJ-induced groups.

**Figure 5 marinedrugs-17-00051-f005:**
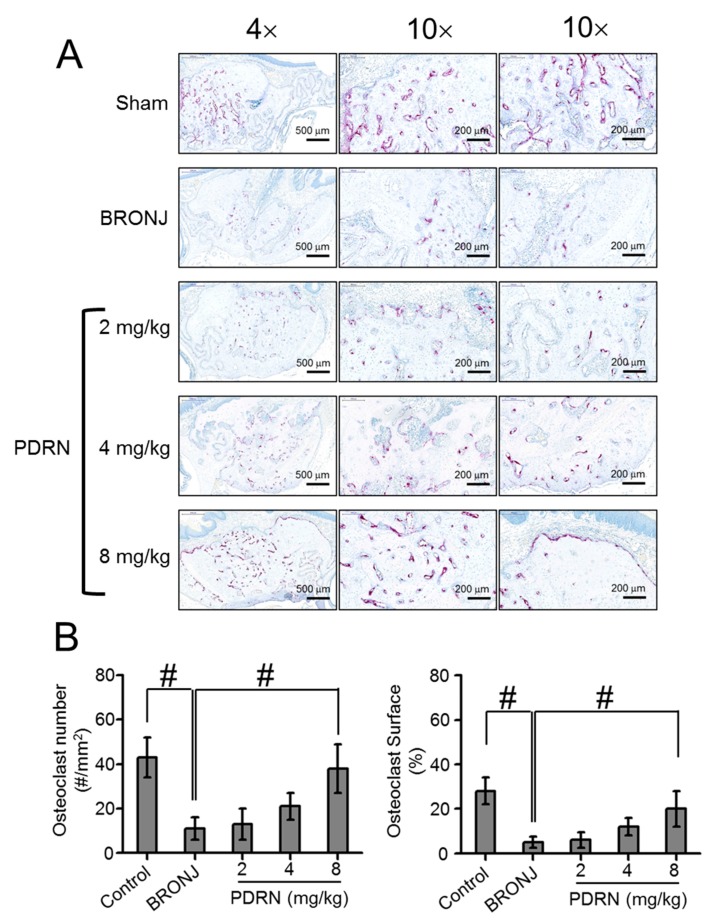
Effect of local PDRN injection on osteoclast number (#/mm^2^) and osteoclast surface (%) at tooth extracted sites. (**A**) Representative TRAP stained images around tooth extracted sites of control (Sharm), BRONJ-induced, and three concentrations of PDRN (2 mg/kg, 4 mg/kg, and 8 mg/kg) treated rats after animal test for 280 days. The osteoclast number and osteoclast surface were observed at a magnification of 4× and 10×. The scale bars at 4× and 10× indicates 500 μm and 200 μm, respectively. (**B**) Osteoclast number and osteoclast surface (%) were measured by calculating the number and percentage per mm^2^, respectively. Statistical analysis was carried out using one-way analysis of variance (ANOVA) (n = 6 rats per group; ^#^
*p* < 0.05). The results of PDRN-treated groups were compared to those of control and BRONJ-induced groups.

**Figure 6 marinedrugs-17-00051-f006:**
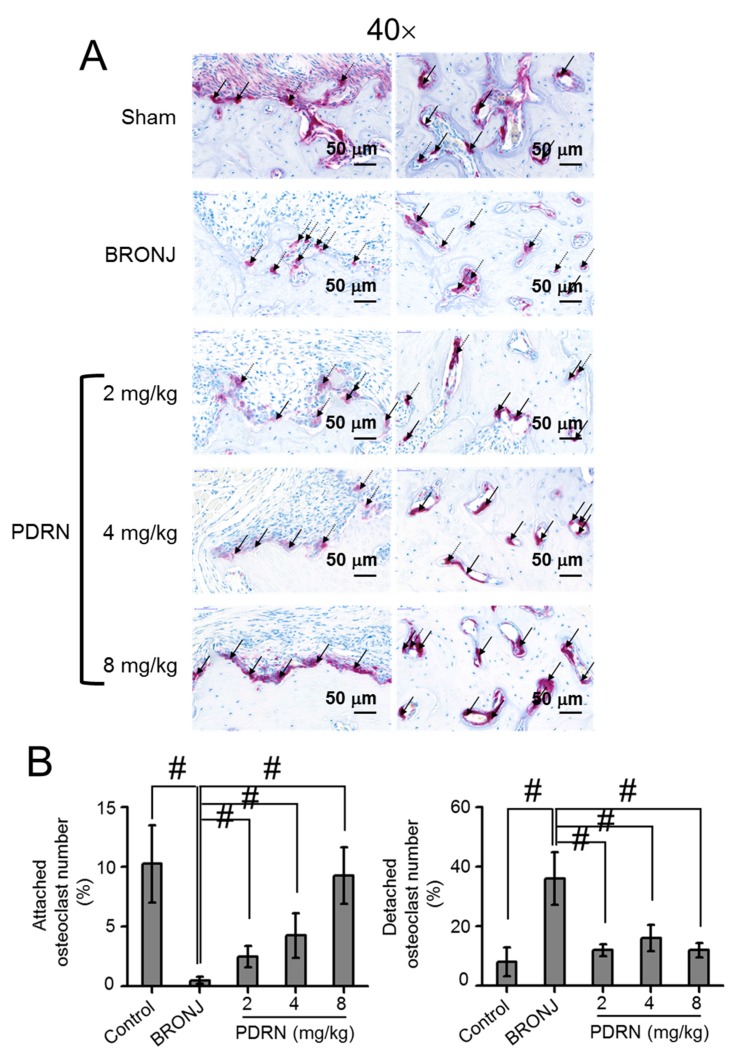
Effect of local PDRN injection on the attached and detached osteoclast numbers (%) at tooth extracted sites. (**A**) Representative TRAP stained images around tooth extracted sites of control (Sharm), BRONJ-induced, and three concentrations of PDRN (2 mg/kg, 4 mg/kg, and 8 mg/kg) treated rats after animal test for 280 days. The attached and detached osteoclast numbers were observed at a magnification of 40×. Black and red arrows indicated attached and detached osteoclasts, respectively. The scale bar indicates 50 μm. (**B**) Attached and detached osteoclast numbers were measured by calculating the percentage per mm^2^. Statistical analysis was carried out using one-way analysis of variance (ANOVA) (n = 6 rats per group; ^#^
*p* < 0.05). The results of PDRN-treated groups were compared to those of control and BRONJ-induced groups.

**Figure 7 marinedrugs-17-00051-f007:**
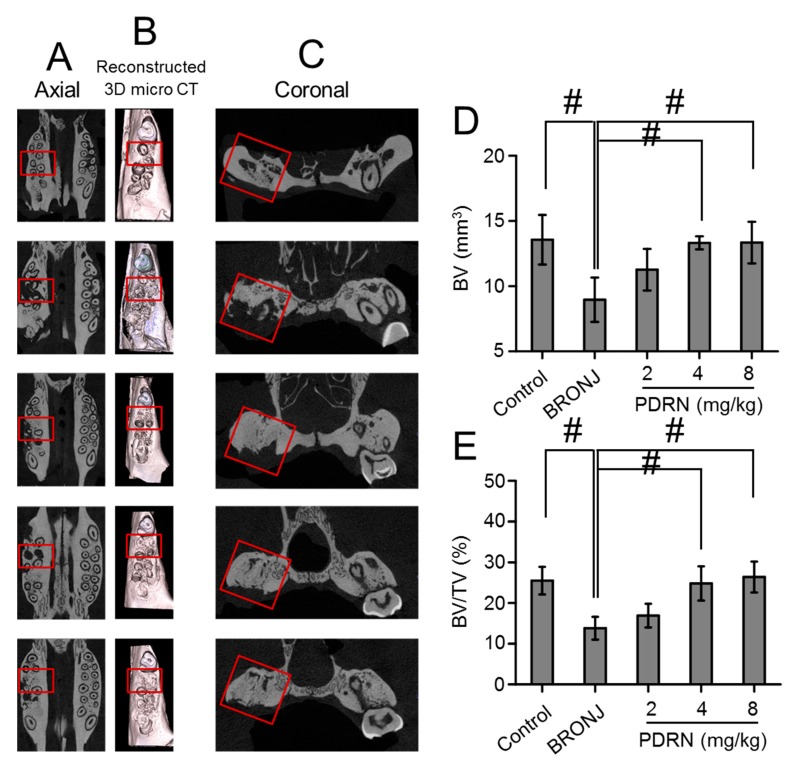
Effect of local PDRN injection on change of bone volume at tooth extracted sites. (**A**) Representative axial micro CT images of tissues around tooth extracted sites of control (Sharm), BRONJ-induced, and three concentrations of PDRN (2 mg/kg, 4 mg/kg and 8 mg/kg) treated rats after animal test for 280 days. (**B**) 3D micro CT reconstructions of the representative images. (**C**) Representative coronal images of tissues around the tooth extracted sites. (**D**) Bone volume (BV, mm^3^) of tooth-extracted sites and (**E**) bone volume/tissue volume (BV/TV, %) determined by micro CT (n = 6 per group). Red squares indicate tooth-extracted sites. Statistical analysis was carried out using one-way analysis of variance (ANOVA) (^#^
*p* < 0.05). The results of PDRN-treated groups were compared to those of control and BRONJ-induced groups.

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
