# Peer review of "The Effect of Polydeoxyribonucleotide Extracted from Salmon Sperm on the Restoration of Bisphosphonate-Related Osteonecrosis of the Jaw"

_marinedrugs, 2019, doi:10.3390/md17010051_

Round 1

Reviewer 1 Report

I read with interest this paper on the benefit of PDRN on the treatment of jaw osteonecrosis in a rabbit model of BRONJ-induced osteonecrosis. The paper provides potential evidence about the benefit of PDRN but there are important methodological issues that need to be resolved before its suitable for publication:

1. how many histological sections from each sample were used for the statistical analysis. The results may be deceiving because of the selection of sections. you need to use several sections from each animal/sample to reach such conclusions

2. the bone volume results need to be reported as bone volume/total volume and the total volume of the defect has to be predefined.

3. which post-hoc test was used for the stats?

4. vessel formation at the periphery of the lesions is common feature of osteonecrosis and does not necessarily mean healing. I can see in the images that the vessels you have identified are indeed in the periphery. Therefore the results on vessel formation may be compromised.

5. in the discussion there should be reference to other models of osteonecrosis and how the model presented here resembles compares to them

6. There is no explanation about the injection of PDRN. How did you make sure that PDRN was delivered at the lesion? I cannot visualise how you injected PDRN in the bone or how it was absorbed to have an effect. Unless a clear explanation is provided to this it is hard to understand why it would have a beneficial effect.

7. Apart from PDRN I would like to see the effects of the local injection of saline or other neutral solution to judge if the effect seen can be attributed to PDRN. Please provide such results.

8. Please provide the statistical significance of all groups compared to the control (after you add the groups asked in comment 7)

9. Please provide more info about the drug in the introduction. Why use it for osteonecrosis of the jaw? You have this information in the discussion so please move it from the discussion to the introduction and expand.

Author Response

Dear Reviewer #1

The authors appreciate valuable comments. We made responses on the comments as follows:

1. how many histological sections from each sample were used for the statistical analysis. The results may be deceiving because of the selection of sections. you need to use several sections from each animal/sample to reach such conclusions.

-> As you noted, the sample number per each group for histological observation was already referred to the 4.1. In vivo Animal study and expressed in red letters.

2. the bone volume results need to be reported as bone volume/total volume and the total volume of the defect has to be predefined.

-> Thank you for your good comment. The authors also thought the results of bone volume/total volume before submission. However, since tooth extracted sites do not have constant sizes, relative comparison of bone volume/tissue volume results on all groups may be barely suitable for evaluation new bone formed in the defected sites. Therefore, the authors suggested that measuring bone volume is a requirement for relative comparison on the recovery of BRONJ in all groups.

3. which post-hoc test was used for the stats?

-> As you noted, the statistical analysis was added in the 4.5. Statistical Analysis and expressed as red letters.

4. vessel formation at the periphery of the lesions is common feature of osteonecrosis and does not necessarily mean healing. I can see in the images that the vessels you have identified are indeed in the periphery. Therefore the results on vessel formation may be compromised.

-> We appreciate your valuable comment. Generally, BRONJ is mainly induced in blood vessel-deficient jaw. Therefore, wound healing should be considered because tooth extractions become triggerpoint. So, blood vessels-deficient bone sites have high incidence rate and blood vessel formation plays an important role in BRONJ recovery. These facts are well explained a reference as follow and the authors extracted significant factors from the referecnes:

Ref) Oncology Reviews, 2014, 8, 254

“Another characteristic that distinguishes the jaw from other bones of the human skeleton is the type of ossification. The maxilla and mandible have an intramembranous ossification, unlike long bones and vertebrae, which have an endochondral ossification. Themandible is denser than any other bone in the human body. Its thickest section, where few vessels are present, is in the premolar and molar region, a site which is usually prone to BRONJ. Additionally, the jaw contains in general a fatty marrow and, in the presence of a hematopoietic environment, it lacks protection during the healing response of the bone.

Various mechanisms of BRONJ have been proposed, but its etiology is not yet fully understood. In order to explain the inadequate healing of wounds, which is the hallmark of BRONJ, some authors have focused on bone tissues, whereas others on soft tissues. To understand the basis of each theory, since most cases of BRONJ have, as trigger point, tooth extractions, it is necessary to consider the wound healing process. The blood clot forms within the first 24 h and is replaced by vascularized granulation tissue in following three days. By the seventh day, the provisional matrix is comprised of new blood vessels and collagen fibers. This is a vulnerable step, if the action of an agent inhibits osteoclasts, when in the adjacent bone marrow spaces osteoclasts should increase in number. By the 14th day, large amounts of new woven bone are found, as well as newly formed blood vessels. Approximately one month after, this woven bone should remodel with an increased osteoclast activity and in the third month after tooth extraction it should be replaced by lamellar bone. This site only contains the bone marrow with lamellar bone after six months. As a result, an impaired osteoclast function could hamper either the early remodeling of the old lamellar bone or the late remodeling of the new bone. Given these mechanisms, it is understandable that the primary lesion lies in the bone. Although, it is unclear why this lesion presents with a loss of soft tissue covering the maxillary bone as its primary clinical feature. This loss may be explained by the fact that BPs accumulate in bone at a high enough concentrations to be directly toxic for the oral epithelium. The lack of healing of the soft tissue lesions after an invasive dental procedure or trauma from dentures lead to secondary infection of the underlying bone, thus worsening the primary lesion.”

5. in the discussion there should be reference to other models of osteonecrosis and how the model presented here resembles compares to them.

-> Rat model have been used for BRONJ-related studies, because the animal have some merits of ease of care and handling, high reproduction, completed genome mapping. Other animal models for BRONJ studies have been little reported; therefore, the authors explained in the Discussion section the reason why the rat model is used for this study.

6. There is no explanation about the injection of PDRN. How did you make sure that PDRN was delivered at the lesion? I cannot visualise how you injected PDRN in the bone or how it was absorbed to have an effect. Unless a clear explanation is provided to this it is hard to understand why it would have a beneficial effect.

-> Thank you for your comment. The explanation was added to the 4.1. In Vivo Animal Study as noted. Since PDRN cannot be directly injected to the defected bone tissue, it was injected to soft tissue near to the hard bone tissue. After the treatment, BRONJ recovery was confirmed in the defected sites; therefore, these results should be attributed to the PDRN treatment.

7. Apart from PDRN I would like to see the effects of the local injection of saline or other neutral solution to judge if the effect seen can be attributed to PDRN. Please provide such results.

-> This study is to firstly confirm the effect of PDRN on BRONJ recovery. Therefore, our first design on this study was to evaluate the results with and without PDRN treatment. Almost 1 year took for this study. Our next study is to conduct tissue-engineered based BRONJ recovery using biomaterials and PDRN. So, your suggestion will be carried out for next study. Please understand time limitation for carrying out your suggestion.

8. Please provide the statistical significance of all groups compared to the control (after you add the groups asked in comment 7).

-> As explained in Response of Comment 7, the suggestion will be carried out as next advanced study using biomaterials and PDRN. Therefore, the authors hope to understand this present situation. 

9. Please provide more info about the drug in the introduction. Why use it for osteonecrosis of the jaw? You have this information in the discussion so please move it from the discussion to the introduction and expand.

-> Thank you for your comment. The information of PDRN was explained in the Discussion section because the use of PDRN has to be clearly discussed with BRONJ recovery. In the Introduction, the simple reason why PDRN is used for PDRN recovery was explained to give motivation for this study. Please understand the author’s intention.

Reviewer 2 Report

The manuscript proposed by Deok-Won et al deals with the possiblity to apply PDRN (at different dosages) to counteract the BRONJ progression that most of the time occurs after bisphosphonates application. Several aspects related to the BRONJ outcome (mucosal coverage, blood vessels formation, bone and osteoclasts repopulation) have been studied by a rabbit model by different techniques (histology mainly).

The manuscript is well written; the methods, including surgery, are properly described as well as results are well presented even if the use of specific markers in combination with classic H&E would have been useful to better detail obtained results.

My only question is related to results interpretation: the Authors used 2-4-8 mg/kg PDRN dosages obtaining sometimes a different response. For example, mucosal and blood vessels formation seems to be improved by a low dosage (2>8 mg/kg, fig 2 and 4) while a higher amount (8 mg/kg) was effective in enhancing both bone (fig 7) and osteoclasts (fig 6) repopulation which is a bit confusing as they come from the same pathway of bone resorption/formation.

So, how the Authors correlate the dosage to the obtained results? Which is the recommended amount for a possible application? I think that those aspects should be discussed to clarify the applicability of PDRN.

Author Response

Dear Reviewer #2

The authors appreciate valuable comments. We made responses on the comments as follows:

The manuscript proposed by Deok-Won et al deals with the possiblity to apply PDRN (at different dosages) to counteract the BRONJ progression that most of the time occurs after bisphosphonates application. Several aspects related to the BRONJ outcome (mucosal coverage, blood vessels formation, bone and osteoclasts repopulation) have been studied by a rabbit model by different techniques (histology mainly).

The manuscript is well written; the methods, including surgery, are properly described as well as results are well presented even if the use of specific markers in combination with classic H&E would have been useful to better detail obtained results.

My only question is related to results interpretation: the Authors used 2-4-8 mg/kg PDRN dosages obtaining sometimes a different response. For example, mucosal and blood vessels formation seems to be improved by a low dosage (2>8 mg/kg, fig 2 and 4) while a higher amount (8 mg/kg) was effective in enhancing both bone (fig 7) and osteoclasts (fig 6) repopulation which is a bit confusing as they come from the same pathway of bone resorption/formation.

So, how the Authors correlate the dosage to the obtained results? Which is the recommended amount for a possible application? I think that those aspects should be discussed to clarify the applicability of PDRN.

-> Thank you for your comment. During bone remodeling, some unique properties, such as necrotic bone formation, empty lacunae formation, blood vessel formation, attached/detached osteoclast formation and new bone formation, are observed in tooth-extracted sites. Therefore, Bone remodeling in tooth-extracted sites can be not explained by a few factors (for example, mucosal coverage and blood vessels). Although a local treatment of 2 mg/kg PDRN had similar blood vessel formation compared with that of 8 mg/kg PDRN, the former resulted in insufficient mucosal coverage. In Fig. 3, larger amount of empty lacunae was observed in a local treatment of 2 mg/kg than 8 mg/kg, indicating possibility of further osteonecrosis. In Figs. 5 and 6, osteoclast formation, in particular attached osteoclasts, plays an important role in bone formation between bone remodeling is a balance between osteoblasts and osteoclasts. After happening bone defect, osteoclasts first acts to remove bone debris at the defected sites and afterward, osteoblasts initiates bone formation at the defected sites. Therefore, considering all data, we believe that our results and discussion are sufficiently persuasive.

Reviewer 3 Report

This manuscript is well done study, but needs additional clarifications, as well as detailed description of the quantification of histological slides. I suggest that the authors address the following issues for clarity and unambiguity

1. Fig 1A shows the time course of treatment. It is not clear why Dexamethasone was administered between day 166 and 180. There is no mention in the text why DEX treatment was necessary. This needs explanation.

2. In Fig 3 the histology of empty lacunae is shown and the lacunae are indicated by arrows. Not all empty lacunae are indicated, just some. How was determined which lacunae to mark by arrows? Just by looking at the slides I see not much difference between the number of lacunae in different slides. Was marking of the lacunae done arbitrary or some computer analysis of the slides was performed? If so, the quantification of histological slides must be described in full detail in Material and Methods. Saying that a particular software was used is insufficient. How was the software used and how was the analysis done; detailed description is needed.

3. The same pertains to Fig 5 where alkaline phosphatase staining is shown. At 10x magnification I see at least two fold difference in staining between sham and 8 mg/kg of PDRN, while quantification in Fig 5B shows no difference. How was the quantification done? Detailed description is needed.

4. No information was provided on the composition and source of PDRN. What was the size range of polynucleotides? What was the source? Detailed description of the PDRN is need in Material and Methods.

Author Response

This manuscript is well done study, but needs additional clarifications, as well as detailed description of the quantification of histological slides. I suggest that the authors address the following issues for clarity and unambiguity

1. Fig 1A shows the time course of treatment. It is not clear why Dexamethasone was administered between day 166 and 180. There is no mention in the text why DEX treatment was necessary. This needs explanation.

Thank you for your comment. The administration of dexamethasone can accelerate the occurrence of BRONJ. This was written in 4.1. Section.

2. In Fig 3 the histology of empty lacunae is shown and the lacunae are indicated by arrows. Not all empty lacunae are indicated, just some. How was determined which lacunae to mark by arrows? Just by looking at the slides I see not much difference between the number of lacunae in different slides. Was marking of the lacunae done arbitrary or some computer analysis of the slides was performed? If so, the quantification of histological slides must be described in full detail in Material and Methods. Saying that a particular software was used is insufficient. How was the software used and how was the analysis done; detailed description is needed.

Empty lacunae indicate spacers without osteoblasts. Empty lacunae were largely distributed in slides; therefore, marking all empty lacunae is difficult. The reason why the authors marked a few empty lacunae was to inform subscribers their existences. The authors wish that the reviewer understand what they intend to do. The number of empty lacunae was determined by calculating its number per 1 mm2. This information was written in 4.3. Section.

3. The same pertains to Fig 5 where alkaline phosphatase staining is shown. At 10x magnification I see at least two fold difference in staining between sham and 8 mg/kg of PDRN, while quantification in Fig 5B shows no difference. How was the quantification done? Detailed description is needed.

We appreciate your valuable comment. As you noted, the information was written in 4.3. Section.

4. No information was provided on the composition and source of PDRN. What was the size range of polynucleotides? What was the source? Detailed description of the PDRN is need in Material and Methods.

We appreciate your valuable comment. As you noted, the information was written in Introduction and 4.1. Section.

Round 2

Reviewer 1 Report

Unfortunately, authors failed to satisfactorily correct most of my previous comments.

Most red sections highlighted in the manuscript were exactly the same as in the original version.

Somehow a new author name appeared during the revision process. This is generally acceptable if extensive revision is required (extensive text changes or new experiments). As proved by the result, authors did not significantly improve any text section based on previous reviewer's suggestions.

Author Response

I read with interest this paper on the benefit of PDRN on the treatment of jaw osteonecrosis in a rabbit model of BRONJ-induced osteonecrosis. The paper provides potential evidence about the benefit of PDRN but there are important methodological issues that need to be resolved before its suitable for publication:

1. How many histological sections from each sample were used for the statistical analysis? The results may be deceiving because of the selection of sections. You need to use several sections from each animal/sample to reach such conclusions.

We appreciate your valuable comment. The section number was expressed in 4.1. Section.

2. The bone volume results need to be reported as bone volume/total volume and the total volume of the defect has to be predefined.

The authors added BV/TV as you noted. Also, the BV in the manuscript indicates the total bone volume formed in the defected sites.

3. Which post-hoc test was used for the stats?

As you noted, the statistical analysis method was expressed in the 4.5.

4. Vessel formation at the periphery of the lesions is common feature of osteonecrosis and does not necessarily mean healing. I can see in the images that the vessels you have identified are indeed in the periphery. Therefore the results on vessel formation may be compromised.

We appreciate your valuable comment. The authors got valuable explanation on this comment from the expert of BRONJ therapy, and it is as follows; Generally, BRONJ is mainly induced in blood vessel-deficient jaw. Therefore, wound healing should be considered because tooth extractions become trigger-point. So, blood vessels-deficient bone sites have high incidence rate and blood vessel formation plays an important role in BRONJ recovery. The expert recommends a reference for resolving this comment as bellows:

Ref) Oncology Reviews, 2014, 8, 254

The below paragraphs are well explaining the importance of blood vessel formation for BRONJ therapy

“Another characteristic that distinguishes the jaw from other bones of the human skeleton is the type of ossification. The maxilla and mandible have an intramembranous ossification, unlike long bones and vertebrae, which have an endochondral ossification. The mandible is denser than any other bone in the human body. Its thickest section, where few vessels are present, is in the premolar and molar region, a site which is usually prone to BRONJ. Additionally, the jaw contains in general a fatty marrow and, in the presence of a hematopoietic environment, it lacks protection during the healing response of the bone.

Various mechanisms of BRONJ have been proposed, but its etiology is not yet fully understood. In order to explain the inadequate healing of wounds, which is the hallmark of BRONJ, some authors have focused on bone tissues, whereas others on soft tissues. To understand the basis of each theory, since most cases of BRONJ have, as trigger point, tooth extractions, it is necessary to consider the wound healing process. The blood clot forms within the first 24 h and is replaced by vascularized granulation tissue in following three days. By the seventh day, the provisional matrix is comprised of new blood vessels and collagen fibers. This is a vulnerable step, if the action of an agent inhibits osteoclasts, when in the adjacent bone marrow spaces osteoclasts should increase in number. By the 14th day, large amounts of new woven bone are found, as well as newly formed blood vessels. Approximately one month after, this woven bone should remodel with an increased osteoclast activity and in the third month after tooth extraction it should be replaced by lamellar bone. This site only contains the bone marrow with lamellar bone after six months. As a result, an impaired osteoclast function could hamper either the early remodeling of the old lamellar bone or the late remodeling of the new bone. Given these mechanisms, it is understandable that the primary lesion lies in the bone. This loss may be explained by the fact that BPs accumulate in bone at a high enough concentrations to be directly toxic for the oral epithelium. The lack of healing of the soft tissue lesions after an invasive dental procedure or trauma from dentures lead to secondary infection of the underlying bone, thus worsening the primary lesion.”

Based on these explanations, the authors discussed angiogenesis and normal vascularization on BRONJ therapy.

5. In the discussion there should be reference to other models of osteonecrosis and how the model presented here resembles compares to them.

Rat model have been used for BRONJ-related studies, because the animal have some merits of ease of care and handling, high reproduction, completed genome mapping. Other animal models for BRONJ studies have been little reported; therefore, the authors explained in the Discussion section the reason why the rat model is used for this study.

6. There is no explanation about the injection of PDRN. How did you make sure that PDRN was delivered at the lesion? I cannot visualize how you injected PDRN in the bone or how it was absorbed to have an effect. Unless a clear explanation is provided to this it is hard to understand why it would have a beneficial effect.

Since PDRN cannot be directly injected to the defected bone tissue, it was injected to soft tissue near to the hard bone tissue. It was injected to soft tissues near to the defected sites. This explanation was written in 4.1. Section.

7. Apart from PDRN I would like to see the effects of the local injection of saline or other neutral solution to judge if the effect seen can be attributed to PDRN. Please provide such results.

This study is to firstly confirm the effect of PDRN on BRONJ recovery. Therefore, our first design on this study was to evaluate the results with and without PDRN treatment. Almost 1 year took for this study. Our next study is to conduct tissue-engineered based BRONJ recovery using biomaterials and PDRN. So, your suggestion will be carried out for next study. Please understand time limitation for carrying out your suggestion.

8. Please provide the statistical significance of all groups compared to the control (after you add the groups asked in comment 7).

As explained in Response of Comment 7, the suggestion will be carried out as next advanced study using biomaterials and PDRN. Therefore, the authors hope to understand this present situation.

9. Please provide more info about the drug in the introduction. Why use it for osteonecrosis of the jaw? You have this information in the discussion so please move it from the discussion to the introduction and expand.

As you noted, the information was moved to Introduction.